# Characterisation of the Paternal Influence on Intergenerational Offspring Cardiac and Brain Lipid Homeostasis in Mice

**DOI:** 10.3390/ijms24031814

**Published:** 2023-01-17

**Authors:** Samuel Furse, Hannah L. Morgan, Albert Koulman, Adam J. Watkins

**Affiliations:** 1Biological Chemistry Group, Jodrell Laboratory, RBG Kew, Kew Road, Richmond, Surrey TW9 3AB, UK; 2Core Metabolomics and Lipidomics Laboratory, Wellcome-MRC Institute of Metabolic Science, University of Cambridge, Addenbrooke’s Treatment Centre, Keith Day Road, Cambridge CB2 0QQ, UK; 3Wellcome-MRC Institute of Metabolic Science and Medical Research Council Metabolic Diseases Unit, University of Cambridge, Cambridge CB2 0QQ, UK; 4Lifespan and Population Health, School of Medicine, University of Nottingham, Nottingham NG7 2UH, UK

**Keywords:** developmental programming, metabolic disorder, paternal diet, semen quality, lipid homeostasis

## Abstract

There is growing evidence that poor paternal diet at the time of conception increase the risk of offspring developing a range of non-communicable metabolic diseases, such as obesity, diabetes and cardiovascular disease, in adulthood. We hypothesise that a paternal low protein–high carbohydrate diet perturbs offspring tissue lipid abundance through both sperm and seminal plasma-mediated mechanisms. To test our hypothesis, we fed male C57BL/6 mice either a control normal protein diet (NPD; 18% protein) or an isocaloric low protein diet (LPD; 9% protein) for a minimum of 8 weeks. We generated offspring through artificial insemination, in combination with vasectomised male mating. Using this approach, we derived offspring from either NPD or LPD sperm but in the presence of NPD or LPD seminal plasma. Using high resolution mass-spectrometry, we found that offspring derived from either LPD sperm or seminal fluid displayed perturbed cardiac and brain lipid abundance from just three weeks of age, typically associated with the altered abundance of tissue triglycerides. We also observed the differential sex-specific patterns of lipids between the control and experimental offspring’s hearts and brains. These observations indicate that poor paternal diet at the time of conception affects offspring cardiac and brain lipid profiles in an age-, sex- and generation-specific manner.

## 1. Introduction

Perturbations in central and peripheral lipid metabolism underlie the aetiology of many conditions, including cardiovascular disease, obesity and type 2 diabetes, where it plays a central role in perturbed tissue metabolism, inflammation, glucose transport and immune cell modulation [1]. Fundamental metabolic dysregulation forms a central component of the metabolic syndrome, a condition linked with increased risk for developing cardiovascular disease, stroke and atherosclerosis [2]. While the development of such life-shortening non-communicable diseases have been attributed to adult lifestyle factors, such as poor diet, smoking and a lack of exercise, studies have identified strong connections between patterns of development in the womb, early neonatal growth and an increased propensity for developing metabolic syndromes and associated comorbidities [3]. Children born premature have been identified as being at higher risk for developing lipid disorders in adult life [4,5], while children born small but gain weight more rapidly during the first three months of life display reduced insulin sensitivity, elevated triglyceride levels and obesity in early adulthood [6]. Similarly, high weight at birth has also been associated with an increased risk of obesity in adulthood [7]. In animal models, a range of sub-optimal maternal diets before, during and after pregnancy have been shown to predispose offspring to becoming overweight and insulin/glucose intolerant and displaying aspects of cardiovascular disease, such as vascular dysfunction [8,9]. Interestingly, studies have also shown altered offspring cognitive and behavioural functions in adult offspring from dietary-manipulated females [10,11,12], indicating that brain development also appears susceptible to dietary perturbations.

Mirroring the association between poor maternal diet and offspring ill-health, studies have identified paternal physiological and metabolic status at the time of conception as a critical factor in programming the health of the offspring [13]. In men, paternal weight at birth, in infants both small and large for their gestational age, has been linked to a decreased or increased birth weight, respectively, of their children [14,15]. Furthermore, paternal obesity has been associated with increased adiposity, triglyceride, insulin, C-peptide and blood pressure in children [16,17] when compared to men of normal weight. Similarly, children of men who have developed type 2 diabetes are at increased risk of developing elevated triglycerides, increased BMI and type 2 diabetes themselves [18,19]. Interestingly, paternal factors, such as advancing age in men, have also been linked to psychiatric morbidities and behavioural changes in their children [20]. In animals, associations between paternal high fat [21], low protein [22,23] and low folate [24] and perturbed offspring growth and cardio-metabolic health have been demonstrated in rodent models.

While the link between poor paternal diet at the time of conception and perturbed development and well-being of adult offspring is gradually being established, the underlying mechanisms remain to be fully defined. Studies in animal models have indicated that both the sperm and the seminal plasma play a role in influencing post-fertilisation development [25]. Factors such as the integrity of the sperm DNA in conjunction with epigenetic modifications (DNA/RNA methylation, histone modifications, non-coding RNAs) have all been identified as sperm-specific modes of offspring programming [26]. Separately, the seminal plasma and the range of paternal antigens, signalling molecules and extracellular vesicles contained within can modulate maternal reproductive tract physiology, gene/protein expression and immune cell populations and responses, all essential for supporting post-fertilisation development [27]. Our own studies have shown that offspring growth, metabolism and cardiovascular function can be perturbed through both sperm and seminal plasma mechanisms in response to a sub-optimal paternal low protein diet (LPD) in mice [28,29,30,31]. In the current study, we tested the hypothesis that a poor quality paternal LPD would modify the lipid abundance profiles in offspring heart and brain tissues through both sperm- and seminal plasma-mechanisms. We chose these tissues as poor health can be driven by both acute and chronic imbalances in the brain–heart connection. Furthermore, the function of both tissues is directly influenced by the availability of circulating lipids and related species [32,33].

## 2. Results

### 2.1. F1 Neonatal Lipids

We designed and developed a mouse model in which we combined artificial insemination with vasectomised male mating to study the sperm- and seminal plasma-specific mechanisms in response to a paternal low protein diet [31]. Using this approach, we generated four groups of offspring, termed NN (NPD sperm and NPD seminal plasma), LL (LPD sperm and LPD seminal plasma), NL (NPD sperm and LPD seminal plasma) and LN (LPD sperm and NPD seminal plasma), in order to study their development and well-being over two generations. In the current study, we profiled cardiac and brain tissue lipid abundance to define whether poor paternal diet affected these tissues similarly over successive generations of offspring. At 3 weeks of age, we observed no significant influence of sex on offspring heart weight, ratio of heart to body weight or lipid profiles, as such data for males and females were combined. F1 NL offspring displayed a significantly lower body weight (Figure 1A, *p* < 0.05) and heart weight (Figure 1B, *p* = 0.014) but a significantly elevated heart:body weight ratio (Figure 1C, *p* = 0.019) when compared to F1 NN and LL offspring. The high-resolution mass-spectrometry analysis of offspring hearts detected approximately 340 separate lipid isoforms in positive ionisation mode, comprising cholesteryl esters (CE), ceramides (CER), diglycerides (DG; water-loss product from fragmentation of triglycerides in source), lyso-phosphatidylcholines (LPC), lyso-phosphatidylethanolamine (LPE); monoglycerides (MG), phosphatidylcholines (PC), phosphatidylethanolamine (PE), phosphatidylglycerol (PG), sphingomyelins (SM) and triglycerides (TG) (see Appendix A for full profile). The initial principal component analysis (PCA) of cardiac lipids revealed no separation between experimental groups (Figure 1D). Despite this, LL offspring hearts contained a significantly lower abundance of saturated lipids when compared to NN hearts (Figure 1E, *p* = 0.042) and a higher abundance of mono-unsaturated lipids when compared to NN (*p* = 0.038) and NL (*p* = 0.031) offspring (Figure 1E). The further characterisation of cardiac lipids revealed NL offspring to have the highest abundance of diglycerides and triglycerides but the lowest abundance of phosphatidylcholine species when compared to all other groups (Figure 1F). The analysis of individual cardiac lipids identified 20 with a differential abundance between treatment groups (Figure 1G). Here, NL offspring displayed a higher abundance of diglycerides DG (26:00; 30:02; 36:04; 36:05) when compared to the other groups. Additionally, NL offspring displayed an elevated abundance of sphingomyelins SM (47:00; 48:01) and several triglycerides TG (44:00; 46:02; 54:05; 54:07; 56:05). LN offspring displayed elevated levels of phosphatidylethanolamines PE (40:06; 40:08) while LL, NL and LN offspring displayed a lower abundance of lyso-phosphatidylcholine LPC (20:04), SM (39:02) and TG (44:01) when compared to NN offspring. While no overall effect of offspring sex was observed, with males displaying lower abundance of specific diglycerides, lyso-phosphatidylcholines and monoglycerides, while females displayed a higher abundance of phosphatidylethanolamines, phosphatidylglycerols and sphingomyelins (Figure 1H,I).

In the brain, as in the heart, no overall effect of sex was observed. No difference in total brain weight was observed between groups (Figure 2A); however, NL offspring displayed an elevated brain:body weight ratio when compared to NN and LL offspring (Figure 2B, *p* < 0.05). The analysis of offspring brains detected approximately 440 separate lipids in positive ionisation mode, comprising cholesteryl esters, ceramides, diglycerides, lyso-phosphatidylcholines, lyso-phosphatidylethanolamine; monoglycerides, phosphatidylcholines, phosphatidylethanolamine, phosphatidylglycerol, sphingomyelins and triglycerides (see Appendix A for full profile). The initial principal component analysis (PCA) of brain lipids revealed no separation of treatment groups from each other (Figure 2C). Unlike the heart, no difference in the relative abundance of saturated, mono-unsaturated or poly-unsaturated lipids was observed between treatment groups (Figure 2D). Similarly, the relative abundance of the major lipid classes within the brain appeared to be equivalent between the four groups (Figure 2E). Despite this, an abundance of several individual lipid isoforms were significantly different with LN offspring displaying elevated levels of specific ceramides Cer (39:01), lysoalkylphosphatidylcholine PC-O (37:01), sphingomyelins SM (40:01; 40:02; 42:01; 42:02) and multiple triglycerides when compared to NN, LL and NL offspring (Figure 2F). Interestingly, greater sex-specific differences were observed in the brain than in the hearts with males displaying a lower abundance of multiple ceramides, phosphatidylcholines and triglycerides when compared to females (Figure 2G,H).

### 2.2. F1 Adult Offspring Lipids

To determine whether the differential lipid abundances observed in F1 neonates were sustained into adulthood, we explored cardiac and brain lipid status in adult offspring at 16 weeks of age (see Appendix A, respectively, for full profile). In males, LL, LN and NL offspring were all significantly heavier than NN males (Figure 3A, *p* < 0.05), while NL females were heavier than NN females (Figure 3A, *p* = 0.02). Unlike at 3 weeks of age, no difference in heart weight (Figure 3B) or heart:body weight ratio (Figure 3C) was observed between either male or female offspring. Using the same high-resolution mass-spectrometry analysis as for the neonates, we observed elevated levels of multiple triglycerides in the LL offspring (Figure 3D) while NL offspring displayed an elevated abundance of diglycerides DG (38:04; 38:06; 40:06; 40:07), phosphatidylethanolamines PE (38:06; 40:07) and sphingomyelins SM (40:10; 42:02). Interestingly, LN and NL offspring both displayed a relatively lower abundance of triglycerides when compared to LL and NN offspring (Figure 3D). While principal component analysis revealed no overall separation between experimental groups or sex in total lipid abundance, multiple individual lipids did show a sex-specific difference. We observed several diglycerides (Figure 3E), phosphatidylcholines (Figure 3F), phosphatidylethanolamines (Figure 3G), sphingomyelins (Figure 3H) and triglycerides (Figure 3I) to be different in their abundance between males and females. Interestingly, for the majority of these lipids, males displayed an elevated abundance when compared to females. The separation of these significant cardiac sex-specific lipid differences based on treatment group showed that NN males displayed a higher abundance of diglycerides than NN females (Figure 3J, *p* = 0.004). However, no other group displayed a significant sex-specific difference. For the phosphatidylcholines, no difference was observed between NN males and females; however, LL and NL males displayed significant differences to their respective females (Figure 3K; *p* < 0.05). Similarly, no sex-specific difference in phosphatidylethanolamine abundance was observed between NN males and females (Figure 3L), while a difference was observed between LL, NL and LN male and female offspring (Figure 3L). Similar to the sex-specific differences seen in diglyceride abundance, NN males displayed significantly elevated profiles of sphingomyelins (Figure 3M) and triglycerides (Figure 3N) when compared to NN females (*p* < 0.05). For both lipid classes, LL, NL and LN offspring showed no such sex-specific differences.

Previously, we demonstrated that adult LL, NL and LN offspring displayed significant impairments in vascular function in response to different vasoconstrictors (phenylephrine) and vasodilators (acetylcholine, with or without the addition of the antagonist of nitric oxide synthase Nώ-nitro-Larginine methyl ester (L-NAME) and sodium nitroprusside) when compared to NN offspring [30]. We also previously demonstrated that the serum lipid profile of LL, NL and LN offspring is different to that of NN offspring, displaying a significantly elevated abundance of multiple triglycerides and phosphatidylcholines [29]. As perturbed circulating lipid profiles are a significant risk factor for cardiovascular disease [34], we compared the lipid abundances of cholesterol, diglycerides, phosphatidylcholines, phosphatidylethanolamines, phosphatidylserines, sphingomyelins and triglycerides between the serum and heart with our previously reported vascular responses [30] in the same mice. As no effect of sex was observed in these analyses, data for males and females were combined. In NN mice, we observed significant positive correlation between serum cholesterol (SCHO) and the maximal vasodilatory response to the nitric oxide donor, sodium nitroprusside (SNP max) (Figure 4A; *p* = 0.02). Furthermore, we saw a positive correlation between serum sphingomyelin (SSM) abundance and the maximal vasodilatory response to acetylcholine (ACh Max) (Figure 4A; *p* = 0.049). LL offspring also displayed positive correlations between the abundance of serum cholesterol (SCHO) and sphingomyelin (SSM) and the maximal vasodilatory response to sodium nitroprusside (SNP Max) and acetylcholine (ACh Max), respectively. However, in contrast to NN offspring, serum diglycerides (SDG) and triglycerides (STG) were negatively correlated to the maximal vasodilatory responses to acetylcholine (ACh Max) and sodium nitroprusside (SNP Max) (Figure 4B; *p* < 0.05) in LL mice. Furthermore, we observed positive and negative correlations between the abundance of each lipid class in the heart and the maximal responses to phenylephrine (PE Max) and acetylcholine (ACh Max) (Figure 4B). In NL offspring, serum phosphatidylcholine (SPC) and cholesterol (SCHO) were negatively and positively correlated to the maximal vasodilatory responses to sodium nitroprusside (SNP Max) and acetylcholine when in the presence of L-NAME (ACh L Max), respectively (Figure 4C; *p* < 0.05). In addition, the abundance of sphingomyelins in the heart correlated negatively to the maximal response to acetylcholine when in the presence of L-NAME (ACh L Max) and sodium nitroprusside (SNP Max), while correlating positively to maximal responses to phenylephrine (PE Max) (Figure 4C; *p* < 0.05). Finally, in LN offspring, the serum abundance of cholesterol and sphingomyelins correlated positively to the maximal vasodilatory responses to acetylcholine (ACh Max), acetylcholine when in the presence of L-NAME (ACh L Max) and sodium nitroprusside (SNP Max) (Figure 4D; *p* < 0.05), while serum diglycerides correlated negatively to the maximal response to sodium nitroprusside (SNP Max) (Figure 4D; *p* = 0.035).

We observed significantly elevated brain weight in adult LN males and females when compared to LL males and NN females, respectively (Figure 5A, *p* < 0.05). Furthermore, NL females also displayed an elevated brain weight when compared to NN females (Figure 5A, *p* = 0.01). When expressed as a brain:body weight ratio, differences were only observed in males with LL males displaying a lower ratio when compared to all other groups (Figure 5B, *p* < 0.05). The principal component analysis (PCA) of lipids either in the cerebrum or cerebellum revealed no separation of treatment groups or sexes from each other -. In the cerebrum, only three lipids, Cer (38:01), Cer (44:02) and DG (42:07), displayed significant differences between experimental groups. The abundance of Cer- (38:01) was elevated in LL offspring when compared to all other groups (*p* < 0.01), while Cer (44:02) was also elevated in LL offspring but only compared to NN offspring (*p* = 0.02). The abundance of DG (42:07) only displayed a difference between LL and NL offspring (*p* = 0.029), being elevated in LL offspring. Similarly, only one lipid in the cerebellum, DG (38:04) displayed a significant difference between groups, being reduced in its abundance in LN offspring when compared to NN offspring (*p* = 0.43).

Similar to the adult heart, there were no overall sex-specific differences in lipid abundance (as determined using a PCA) either in the cerebrum or cerebellum. However, in the cerebellum, we observed multiple differences between the sexes when individual lipid abundance was analysed. We observed several ceramides (Figure 5C), diglycerides (Figure 5D), phosphatidylcholines (Figure 5E) and triglycerides (Figure 5F) to be significantly different in their abundance between males and females (*p* < 0.05). As in the heart, for the majority of these lipids, males displayed a higher relative abundance than females. When separated for the different treatment groups, we observed that LL males had a higher abundance of ceramides when compared to LL females (Figure 5G, *p* = 0.32). However, this difference was not present in any of the other groups. For the differential diglycerides, a sex-specific difference was only observed in the NN group, with males possessing a higher abundance than the females (Figure 5H, *p* = 0.033). No individual treatment group displayed any sex-specific differences in their levels of phosphatidylcholines (Figure 5I); however, NN males displayed elevated levels of triglycerides when compared to NN females (Figure 5J, *p* = 0.046), a difference not present in the other groups.

### 2.3. F2 Neonatal Offspring Lipids

We observed that metabolic and cardiovascular ill-health phenotypes are programmed across two generations of offspring in response to poor paternal diet [28,29,30]. To determine whether perturbed cardiac and brain lipid abundance profiles could also be sustained into a second generation, we profiled heart and brain tissues from the F2 offspring (see Appendix A, respectively, for full profile). As no effect of offspring sex was observed at this time, data for males and females were combined. At three weeks of age, we observed that second-generation LL offspring were significantly heavier than NN offspring (Figure 6A, *p* = 0.01). LL and NL offspring displayed a significantly heavier heart weight than NN offspring (Figure 6B, *p* < 0.05). Similarly, brain weight in F2 LL offspring was significantly heavier than for NN offspring (Figure 6D, *p* = 0.036). Despite these differences, when expressed as a ratio to body weight, no difference in heart (Figure 6B) or brain (Figure 6E) weights were observed. The analysis of cardiac lipids revealed only five lipids, DG (26:03), DG (37:00), PE-O (40:05), SM (44:02) and TG (49:01) to be of different abundance between groups. Furthermore, only three lipids, PC-O (40:03), PI (34:02) and TG (38:03) displayed sex-specific differences, with males having a higher abundance than females for all (*p* < 0.05). In the brain, 23 (Figure 6F) and 29 (Figure 6G) lipids displayed different abundances in the cerebellum and cerebrum respectively between treatment groups. In the cerebellum, NL and LN offspring displayed an elevated abundance of several saturated and unsaturated phosphatidylcholines when compared to NN and LL offspring, while also displaying a lower abundance of phosphatidylcholine plasmalogen PC-O (33:02; 38:01; 40:03) and ceramides Cer- (41:01; 41:02; 44:03) (Figure 6F). In the cerebrum, LL offspring displayed elevated abundance of multiple ceramides, phosphatidylcholines and triglycerides (Figure 6G). Interestingly, NL offspring displayed an abundance profile similar to that of the LL offspring, while the LN offspring displayed a profile more similar to that of the NN offspring. In line with the F2 generation heart, we observed comparatively few sex-specific differences in lipids in the cerebrum (five lipids; DG (36:05), PCO (40:03), PE (34:00; 36:00; 38:01) and cerebellum (seven lipids; Cer (40:01), PC (34:02; 38:05; 32:00), PI (37:05), PS (44:08), and SM (41:01)). These observations indicate that during early life, lipid abundance in both F1 and F2 offspring are influenced more by their treatment group. However, in the F1 adults, sex becomes the more dominant factor in shaping tissue lipid profiles as the offspring age.

## 3. Discussion

Our previous work showed that a suboptimal, paternal LPD modifies offspring growth and metabolic and cardiovascular well-being, underlined by significant modulations in tissue and serum lipid abundances [28,29,30,31]. While our observations indicated that paternal LPD had little impact on fundamental fertility, we observed that the changes in offspring phenotype were programmed via distinct sperm- and seminal plasma-mediated mechanisms [31]. In the present study, we extended our analysis of the effects of poor paternal diet on offspring health in order to characterise the lipid profiles of offspring heart and brain tissue. We observed that neonatal offspring displayed minimal differences in the lipid abundance in their hearts and brains at 3 weeks of age. However, as the offspring aged, significant sex-specific changes in tissue lipid abundance between groups occurred. Interestingly, in the second-generation offspring, we observed minimal differences in cardiac lipid abundance but a greater impact on brain lipid profiles than those observed in first-generation offspring. These data indicate that poor paternal diet at the time of conception alters offspring tissue lipid abundance in an age-, sex- and generation-dependent manner.

In F1 neonatal offspring, we observed significant changes in the size and relationship of the heart and brain to body weight in NL mice. We have previously reported on litter size parameters in these mice, observing no difference in mean litter size or sex ratio between groups [31]. Furthermore, as all dams were fed the same standard chow diet, differences in maternal dietary composition cannot explain any differences in offspring growth. As such, these changes in neonatal organ weight may be related to different trajectories in foetal development. Previously, we have shown that foetal growth and placental morphology appear altered in response to the same paternal LPD as used in this study [35]. However, no assessment of foetal development was obtained from the current cohort of animals and so the similar assessment of late gestation development cannot be made. Interestingly, the NL treatment group represent mice derived from normal protein diet sperm and low protein diet seminal plasma. The influences of the seminal plasma on early offspring development have been demonstrated in separate studies [27]. In mice, the removal of the seminal vesicles, which contribute the majority of the seminal plasma volume, impairs embryonic development but programmes adult obesity, glucose intolerance and elevated blood pressure [36]. In the pig, infusions of sperm-free seminal plasma during oestrus have been shown to advance ovulation, support corpora luteal development and influence the expression of genes associated with embryo development and implantation within the uterus for up to six days after exposure [37]. In cows, repeated mating to vasectomised bulls, and therefore exposure to seminal plasma, results in longer filamentous conceptuses in comparison to control heifers [38]. These observations highlight the significant role that seminal plasma plays in mediating post-fertilisation development and the maternal environment, which can all affect foetal and neonatal growth.

In the hearts of F1 NL neonates, we observed the elevated abundance of several diglycerides (22:00; 30:02; 36:04; 36:05), sphingomyelins (47:00; 48:01) and triglycerides (44:00; 46:02; 54:05; 54:07; 56:05) when compared to the other groups. The accumulation of metabolic intermediates, such as diglycerides and triglycerides, in the heart has been linked with insulin resistance and a range of cellular perturbations, including apoptosis and endoplasmic reticulum stress and oxidative stress [39]. In skeletal muscle, intramyocellular diglyceride accumulation has been linked with insulin resistance, blunted insulin signalling and lipotoxicity [40]. However, some studies have indicated that streptozotocin-induced diabetic myocardial dysfunction can be improved by increasing the levels of cardiac diglycerides and fatty acids in rats [41]. Furthermore, the size and extent of cardiac damage during ischemia-reperfusion injury in rabbits was attenuated following direct protein kinase C activation [42]. As such, the precise impact and significance of the elevated lipids seen in the F1 NL neonates requires further investigation but could be an indicator of subsequent tissue insulin resistance and dysfunction, as we reported previously [30].

In the brains of the F1 neonates, we observed a significantly higher brain:body weight ratio in NL offspring, mirroring that seen for the heart. However, the analysis of individual lipid abundance revealed significant elevations of a range of sphingomyelins (40:01; 40:02; 42:01; 42:02) and triglycerides (46:05; 47:02; 48:02; 48:04; 48:05; 48:06; 49:03) in LN offspring when compared to the other groups. Interestingly, triglycerides such as these may represent lipids liberated during lipolysis and so may indicate and early association between dysfunctional adipose regulation and brain lipid abundance. Lipids account for approximately 50–60% of the weight of the brain [43] with the myelin sheath comprising a significant number of them. In the brain, lipids are used as signalling molecules and sources of energy, as well as for neurogenesis and synaptogenesis [44]. Many neuronal disorders are underlined by changes in lipid abundance, compartmentalisation or metabolism [45]. Sphingolipids are important in neuronal membrane organization and plasticity. In mice, deficiencies in neutral sphingomyelinase-2 (nSMase), an enzyme which hydrolyses sphingomyelin to ceramide, results in impaired episodic-like and spatial compromised plasticity [46]. In addition, sphingolipids have also been shown to stimulate glutamate release from synaptosomes and enhance hippocampal glutamate transmission rate in mice [47]. While the role of elevated triglycerides has been associated with metabolic and cardiovascular dysfunction, their role in regulating brain health is less well-defined. Studies have made connections between maternal diet in pregnancy and offspring brain composition. The neonatal offspring of pregnant mice fed an n-3 poly-unsaturated fatty acid enriched diet displayed reduced levels of eicosapentaenoic acid, n-3 docosapentaenoic acid and docosahexaenoic acid when compared to non-supplemented offspring [48]. However, all females used in our current study were fed the same standard chow diet during pregnancy and weaning, meaning maternal dietary intake cannot account for the changes in offspring brain lipids. Therefore, the mechanisms altering brain lipid composition in the neonatal offspring remain undefined.

In adult offspring, no difference in heart weight or size relative to body weight was observed. Furthermore, we only observed minimal differences in heart lipid abundance between groups. Here, LL offspring displayed lower and higher abundance of several diglycerides and triglycerides, respectively, while NL offspring displayed an elevated abundance of diglycerides and phosphatidylethanolamines. Interestingly, both NL and LN offspring displayed reduced cardiac triglycerides. As discussed earlier, such differing amounts of cardiac di- and triglycerides could be an indicator of tissue insulin resistance [49]. Our main observation in the F1 adult heart and brain tissues was the presence of significant sexual dimorphisms in the abundance of several lipid classes, which were different between groups. In the heart, NN offspring displayed sex-specific differences in the abundance of diglycerides, sphingomyelins and triglycerides, which were not present in the LL, NL or LN offspring. In contrast, sex-specific differences in the abundance of phosphatidylcholine and phosphatidylethanolamine were present in the LL, NL and LN offspring but not in the NN offspring. Many studies have shown that the lipoprotein profile of females is different to that of males, with men displaying higher levels of circulating triglyceride and LDL-cholesterol but lower HDL-cholesterol than women [50]. These differences have been shown to correlate with the well-established higher risk of cardiovascular disease in men [50], though these sex-specific differences diminish with age [51]. In mice, differences in lipid metabolism between the sexes have also been reported [52] as well as being programmed through a maternal high fat diet [53]. In addition, the sexual dimorphic patterns of gene expression have been reported in rodent and human tissues including the heart [54], skeletal muscle [55] and brain [56]. Typically, sex-specific differences are attributed to the role of oestrogen and testosterone in regulating basal metabolic status in females and males, respectively, as well as the different sets of genes located on the X and Y chromosome. As such, sexual dimorphism in tissue function, structure, metabolism and genes expression is to be expected. However, we observe opposite patterns of sexual dimorphic cardiac lipid abundance in our LL, NL and LN offspring when compared to the NN offspring. These observations could indicate differential profiles of oestrogen and testosterone in our experimental animals, changing the normal patterns of downstream signalling from oestrogen, progesterone and/or androgen receptors [57]. Alternatively, the modified sex-specific differences observed within the heart could reflect differential responses to growth hormone (GH). While GH is a major determinant of body growth, it has also been show to regulate hepatic gene expression in mice, influencing central metabolic regulation [58]. In addition, dietary-induced obese mice show sex-specific changes in mitochondrial metabolism with females displaying a greater metabolic flexibility than males [59]. We have previously shown that LL, LN and NL offspring become heavier than NN offspring from approximately 5 weeks of age with underlying glucose intolerance and changes in hepatic metabolic gene expression [31]. These changes also appeared to a greater degree in male offspring than in females. Together, these data indicate that poor paternal diet induces a disruption to the normal sexual-dimorphic profile of the cardiac tissue, programmed potentially through perturbed endocrine regulation and cellular metabolism.

The connection between elevated lipid profiles and cardiovascular disease is now well established. To determine whether the changes in cardiac tissue and serum (as reported previously [29]) lipid abundance seen in our current study related to the changes in vascular function seen in our previous study [30], where we correlated the maximal vascular responses to a range of vasoconstrictors (phenylephrine) and vasodilators (acetylcholine, acetylcholine with L-NAME, sodium nitroprusside). In NN offspring, we observed only two positive associations between the levels of serum cholesterol and sphingomyelin and the maximal vasodilatation to the nitric oxide donor sodium nitroprusside and to acetylcholine with L-NAME, respectively. While elevated levels of serum cholesterol are typically associated with cardiovascular ill-health [60], we observed positive correlations between serum cholesterol and all aspects of vascular function in our NN mice. In LL mice, we observed a more dynamic series of correlations between serum and heart lipid abundance and vascular function. These correlations were mainly associated between vascular constriction in response to phenylephrine and the abundance of lipids in the heart and serum lipids and the vasodilatation responses to sodium nitroprusside. In the LN and NL offspring, fewer associations between serum and cardiac lipids and vascular function were observed than for LL offspring. While we are unable to attribute the precise function of each individual lipid class with vascular function and cardiac well-being, perturbed connections between metabolic status, lipid profiles and cardiovascular health are well established [61]. Elucidating the role that different lipid classes may play in regulating cardiovascular function will provide a clearer understanding of the underlying pathophysiological mechanisms relating programmed metabolic ill-health. Furthermore, studies such as ours provide a wider understanding of which lipid molecules and classes require more detailed focus and investigation for subsequent lipid-modulating therapeutic studies.

Similar to the sex-specific lipid abundances seen in the adult hearts of the offspring, several lipid classes also showed significant sexual dimorphism in the brain. We observed elevated abundances of ceramides, diglycerides, phosphatidylcholine and triglycerides in male brains when compared to females. As with the heart, NN offspring displayed sex-specific differences in the abundance of diglycerides and triglycerides, which were not present in the other groups. In contrast, LL offspring alone displayed a sex-specific difference in the abundance of ceramides. Male mice have been shown to have a higher proportion of fatty acids in their brains than females [62] with male brains containing more saturated fatty acids and females containing more poly-unsaturated fatty acids. These differences in male and female brain lipid composition may be derived from differential uptake, turnover or de novo synthesis between the sexes. Sex-specific differences in the expression of fatty acid receptors in skeletal muscle have been reported [63]. Furthermore, the expression of the proteins involved in the binding and transport of fatty acids, including fatty acid binding proteins, CD36 and long-chain fatty acid transport protein 4 (Slc27a4) are higher in women than in men [63]. In rats, the expression of the hepatic elongation of very long chain fatty acids protein 6 (Elovl6) involved in the synthesis of unsaturated fatty acids is higher in females than males [64], potentially contributing to differences in saturated fats seen between males and females. While these differences in the brain were not as pronounced as those seen in the heart, they do indicate that the lipid composition of the brain is also influenced by the dietary background of the father.

Our final observation was that while few differences in lipid abundance were observed in the hearts of our second-generation offspring, multiple ceramides, phosphatidylcholines and triglycerides displayed differential abundance in the brain. We have shown that offspring cardiovascular [30] and metabolic [29] phenotype are impaired over two generations in response to poor paternal diet. Separate studies have also shown that paternal obesity at the time of mating impairs offspring growth [65], glucose tolerance and hepatic lipid metabolism [66] and reproductive fitness [67] in mice over multiple generations. Interestingly, we observed a greater number of individual lipids to have a differential abundance in F2 neonatal hearts and brains than in our F1 neonates. Previously, we demonstrated that in the livers of the same animals as analysed here, the ratio of phosphatidylcholine to phosphatidylethanolamine (PC:PE) in NL and LN offspring was more similar to the F1 adults than the F1 neonates [29]. While these data indicate a drift in tissue lipid profile across generations, more complex modelling of the lipid abundance between tissues, sexes, ages and generations is required to fully understand intergenerational paternal programming. However, such intergenerational programming effects indicate an underlying epigenetic transmission of paternal environmental traits as a potential mechanism. Differential sperm DNA, RNA, histone and non-coding RNA profiles have been identified in response to a wealth of paternal lifestyle, physiological and environmental perturbations [68] and provide a key mechanism through which a father can influence multiple generations. Additionally, our existing, and previous, data show that the seminal plasma also plays a significant role in influencing offspring phenotype and life-long wellbeing. These seminal plasma effects are likely mediated through responses in the maternal uterine immune, vascular and inflammatory status during the earliest stages of development [27]. Together, these data highlight the mechanistic complexity of paternal programming.

## 4. Materials and Methods

### 4.1. Animal Housing

Male (intact and vasectomised) 8-week-old C57BL/6 male mice (Harlan Ltd., Belton, Leicestershire, UK) were maintained at Aston University’s biomedical research facility in a 07:00–19:00 light–dark cycle at a temperature of 20–22 °C with ad libitum access to chow and water. Weight-matched male mice were housed singly and allocated either a control normal protein diet (NPD; 18% casein; *n* = 16 intact and 8 vasectomised males) or an isocaloric low protein diet (LPD; 9% casein; *n* = 16 intact and 8 vasectomised) for a minimum of 8 weeks. Diets were manufactured and purchased from Special Dietary Services Ltd., (Witham, UK) and formulated to a composition as detailed in Appendix A. Virgin 8-week-old female C57BL/6 mice (Charles River, London, UK) were maintained at Aston University’s biomedical research facility in a 07:00–19:00 light–dark cycle at a temperature of 20–22 °C with ad libitum access to chow and water.

### 4.2. Offspring Generation

Virgin females were super-ovulated with 1IU pregnant mare serum gonadotrophin, followed 48 h later with 1IU human chorionic gonadotrophin (Intervet, Milton Keynes, UK). At 12 h after the human chorionic gonadotrophin injection, mature, motile epididymal sperm were isolated from NPD- and LPD-fed males. Sperm were allowed to swim out and capacitate for 90 min at 37 °C (5% CO_2_ in air) in swim out medium (135 mM NaCl, 5 mM KCl, 1 mM MgSO_4_, 2 mM CaCl_2_, 30mM HEPES; supplemented freshly with 10 mM lactic acid, 1 mM sodium pyruvate, 20 mg/mL BSA, 25 mM NaHCO_3_). Females were non-surgically artificially inseminated with 0.05mL of sperm (approximately 10^7^ sperm) in swim out medium. Immediately following artificial insemination, females were placed overnight with an NPD- or LPD-fed vasectomised male. The presence of a vaginal plug the following morning indicated successful mating. Females were monitored for signs of weight gain, indicating pregnancy. Females who did not become pregnant were removed from the study and were not mated again. All females received standard chow and water ad libitum and were allowed to develop to term with their pregnancies. Our experimental insemination and mating strategy generated four groups of offspring termed ‘NN’ (NPD sperm and NPD seminal plasma), ‘LL’ (LPD sperm and LPD seminal plasma), ‘NL’ (NPD sperm and LPD seminal plasma) and ‘LN’ (LPD sperm and NPD seminal plasma). All offspring received standard chow and water ad libitum throughout the study. For the production of an F2 generation, 16-week-old F1 males (*n* = 6 males per treatment group; each from separate litters) were mated naturally to virgin, 8-week-old female C57BL/6 mice (Charles River, London, UK), which were acquired separately for the purpose of mating with F1 males. All F2 offspring received standard chow and water ad libitum.

### 4.3. Offspring Tissue Collection and Lipid Isolation

Male and female F1 offspring were culled by cervical dislocation at either 3 (neonatal) or 16 (adult) weeks of age. All F2 offspring were culled at 3 weeks of age. Both times, hearts and whole brain tissues were snap frozen and stored at −80 °C. Blood was collected via heart puncture, centrifuged at 10,000 rpm, 4 °C for 10 min and the serum was aliquoted and stored at −80°.

Purified internal lipid standards were purchased from Avanti Polar lipids Inc. (Alabaster, Alabama, US). Fine chemicals and solvents were purchased from Sigma-Aldrich (Gillingham, Dorset, UK) and not purified further. Plasticware was purchased from Sarstedt (Leicester, Leicestershire, UK) Chromacol (Leominster, MA, USA) or Integra (Bath, Somerset, UK). Tissues were prepared and extracted as described recently [69]. Solutions of homogenized tissue preparations were injected into a well (96-well plate, Esslab Plate+™, 2·4 mL/well, glass-coated) followed by a methanol solution of internal standards (150 µL), water (500 µL) and DMT (500 µL) using a 96-channel pipette (Integra Viaflo). The mixture was agitated before being centrifuged (3200× *g*, 2 min). A portion of the organic solution (20 µL) was transferred to a running (high throughput) plate (384 well, glass-coated, Chromacol Esslab Plate+™) before being dried (N_2 (g)_). When enough samples to fill 4 × 96 well plates had been processed, the dried films were re-dissolved (*tert*-butylmethyl ether, 20 µL/well; MS-mix, 80 µL/well) and the plate was heat-sealed and run immediately, with the first injection within 10 min.

### 4.4. Offspring Tissue Lipidome Profiling

Samples were infused into an Exactive Orbitrap (Thermo Fisher, Hemel Hampstead, UK), using a TriVersa NanoMate (Advion, Ithaca, NY, USA) autosampler, for direct infusion mass spectrometry (DI-MS [70,71]). Samples (15 μL ea.) were nanosprayed at 1·2 kV in the positive ion mode. The Exactive Orbitrap started acquiring data 20 s after sample aspiration began and continued for 1 min. The Exactive Orbitrap acquired data with a scan rate of 1 Hz (giving a mass resolution of 100,000 full width at half-maximum at 400 *m*/*z*). Automatic Gain Control was set to 3,000,000 and the maximum ion injection time to 50 ms. After 72 s of acquisition in positive ionisation mode, the NanoMate and Exactive switched over to negative ionisation mode, decreasing the voltage to −1·5 kV and the maximum ion injection time to 50 ms. The spray was maintained for another 66 s, after which the NanoMate and Exactive switched over to negative mode with collision-induced dissociation (CID, 70 eV) for a further 66 s. The spray was then halted and the tip discarded, before the analysis of the next sample began. The plate was kept at 15 °C throughout the acquisition. Samples were run in row order. The instrument was operated in full scan mode from *m*/*z* 150–1200 Da.

The lipid signals obtained are reported as relatively abundant (‘semi-quantitative’), with the signal intensity of each lipid expressed relative to the total lipid signal intensity, for each sample, per mille (‰). Raw high-resolution mass-spectrometry data were processed using XCMS (www.bioconductor.org; access date 10 May 2020) and Peakpicker v 2.0 using an in-house R script [70], as previously described [28]. Search lists of known species (by *m*/*z*) were used for positive, negative and negative-with-CID modes (~8k species). Signals that deviated by more than 9 ppm from the exact mass were discarded, as were those with a signal/noise ratio of <3 and those found in fewer than 50% of samples. The correlation of signal intensity to concentration of the lipid variable in QC samples was used to identify lipid signals that were proportional to abundance in the sample type and volume used (threshold for acceptance was a correlation of >0.75). Signals were then signal corrected (divided by the sum of signals for that sample not including internal standards), in order to be able to compare samples in a manner un-confounded by total lipid mass. All statistical calculations were performed on these finalised values. This method cannot distinguish between different lipids with the same molecular formula. Dual spectroscopy [71] was used to interpret and finalise the lipidomic data. Specifically: the ^31^P NMR data of brain and heart tissue were collected and assigned (according to [72,73,74,75]) and used to determine the difference in ionisation efficiency between species.

### 4.5. Statistical Analysis

Offspring body and organ weights were analysed using a multilevel random effects regression model (SPSS version 23) [31] with the paternal origin of the litter incorporated as a random effect. Interactions between treatment group and offspring sex were defined and, where significant effects of offspring sex were identified, data for each sex were analysed separately and reported as such. The analysis of correlation between variables was conducted using either Spearman or Pearson correlation, depending on whether the data were normally distributed or not. The analysis of tissue lipid profiles was conducted as described previously [28]. A Benjamini–Hochberg false discovery rate analysis was applied to all lipidomic data. Significance was taken at *p* < 0.05.

## 5. Conclusions

Our study extends our understanding of the impacts of poor paternal diet on the metabolic and cardiovascular health of offspring. We demonstrate that the lipid abundance profiles of offspring hearts and brains are influenced by the diet of the father at the time of conception. Furthermore, these influences are mediated through both sperm- and seminal plasma-specific mechanisms and are manifested in tissue-, age-, sex- and generation-specific patterns. Our observations that poor paternal diet modulates the normal sex-specific patterns of tissue lipid abundance offers a new insight into a phenomenon widely reported in many studies of development programming, namely that one sex is affected more than the other [76] Interestingly, in our current study, we show that poor paternal diet removes sexual dimorphisms that should be present or establishes new sexual dimorphisms that should not. However, further studies are needed to explore and define the precise mechanisms linking poor paternal diet at the time of conception with the developmental programming of offspring health.

## Figures and Tables

**Figure 1 ijms-24-01814-f001:**
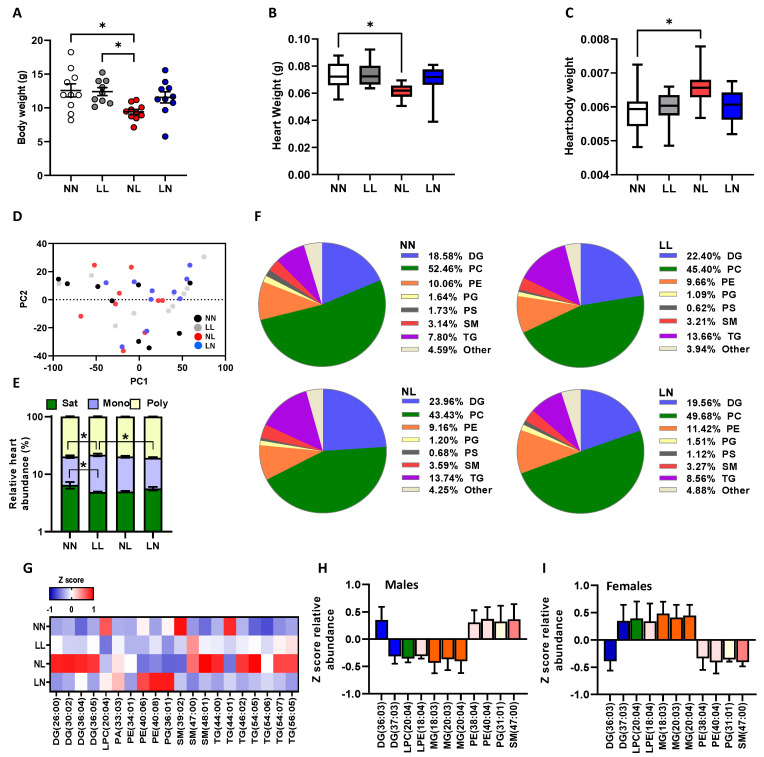
Impact of paternal diet on F1 neonatal offspring heart tissue lipid profiles: (**A**) the body weight, (**B**) heart weight and (**C**) heart:body weight ratio of NN (NPD sperm and LPD seminal plasma), LL (LPD sperm and LPD seminal plasma), NL (NPD sperm and LPD seminal plasma) and LN (LPD sperm and NPD seminal plasma) offspring at 3 weeks of age. (**D**) Principal component analysis (PCA) of differential lipids in NN, LL, NL and LN offspring hearts. (**E**) Relative levels of saturated, mono-unsaturated and poly-unsaturated lipids in NN, LL, NL and LN offspring hearts. (**F**) Relative levels of diglycerides (DG; water-loss product from fragmentation in source), phosphatidylcholines (PC), phosphatidylethanolamines (PE), phosphatidylglycerols (PG), phospatidylserines (PS), sphingomyelins (SM), triglycerides (TG) and other lipids in NN, LL, NL and LN offspring hearts. (**G**) Z scores of differential lipids between NN, LL, NL and LN offspring hearts. Differential lipid abundance in (**H**) male and (**I**) female offspring. N = 9–10 offspring (4–5 males and females) per treatment group, sampled from all litters generated. Data are expressed as mean ± SEM (**A**–**C**). * *p* < 0.05. Statistical differences were determined using a random effects regression analysis (**A**,**B**) or one-way ANOVA with Bonferroni post hoc correction (**E**–**H**).

**Figure 2 ijms-24-01814-f002:**
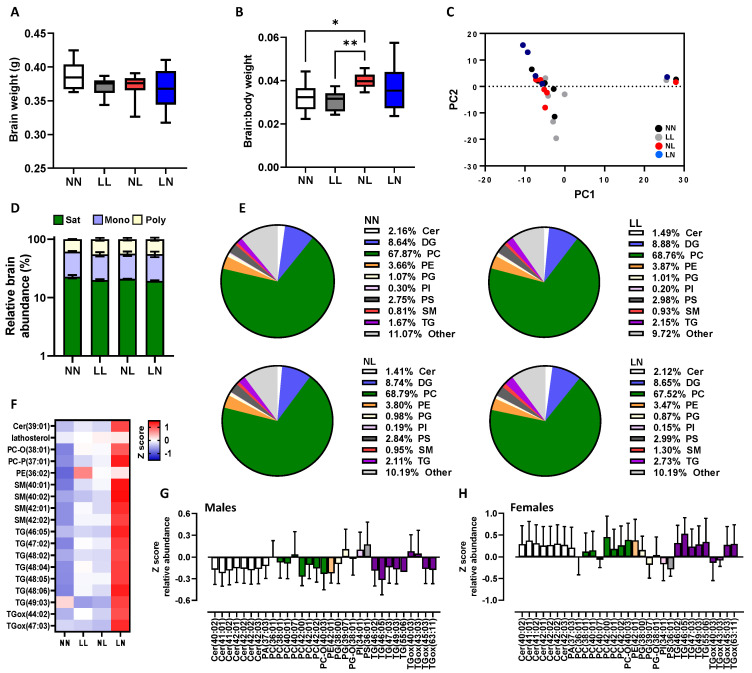
Impact of paternal diet on F1 neonatal offspring brain tissue lipid profiles: (**A**) the brain weight and (**B**) brain:body weight ratio of NN (NPD sperm and LPD seminal plasma), LL (LPD sperm and LPD seminal plasma), NL (NPD sperm and LPD seminal plasma) and LN (LPD sperm and NPD seminal plasma) offspring at 3 weeks of age. (**C**) Principal component analysis (PCA) of differential lipids in NN, LL, NL and LN offspring brains. (**D**) Relative levels of saturated, mono-unsaturated and poly-unsaturated lipids in NN, LL, NL and LN offspring brains. (**E**) Relative levels of ceramides (Cer) diglycerides (DG; water-loss product from fragmentation in source), phosphatidylcholines (PC), phosphatidylethanolamines (PE), phosphatidylglycerols (PG), phosphatidylinositols (PI), phosphatidylserine (PS), sphingomyelins (SM), triglycerides (TG) and other lipids in NN, LL, NL and LN offspring brains. (**F**) Z scores of differential lipids between NN, LL, NL and LN offspring brains. Differential lipid abundance in (**G**) male and (**H**) female offspring. N = 5–6 offspring (2–3 males and females) per treatment group, sampled from all litters generated. Data are expressed as mean ± SEM (**A**,**B**). * *p* < 0.05, ** *p* < 0.01. Statistical differences were determined using a random effects regression analysis (**A**,**B**) or one-way ANOVA with Bonferroni post hoc correction (**D**–**H**).

**Figure 3 ijms-24-01814-f003:**
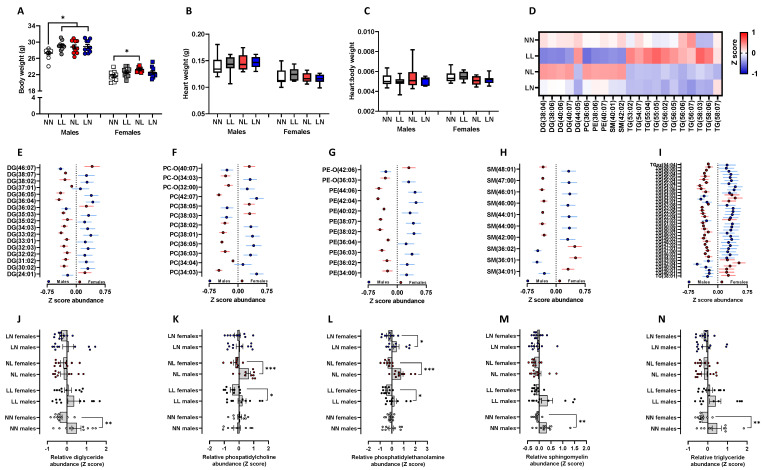
Impact of paternal diet on F1 adult offspring heart tissue lipid profiles: (**A**) the body weight, (**B**) heart weight and (**C**) heart:body weight ratio of NN (NPD sperm and LPD seminal plasma), LL (LPD sperm and LPD seminal plasma), NL (NPD sperm and LPD seminal plasma) and LN (LPD sperm and NPD seminal plasma) offspring at 16 weeks of age. (**D**) Z scores of differential lipids between NN, LL, NL and LN offspring hearts. Sex-specific differences in cardiac (**E**) diglycerides (DG; water-loss product from fragmentation in source), (**F**) phosphatidylcholine (PC), (**G**) phosphatidylethanolamines (PE), (**H**) sphingomyelins (SM) and (**I**) triglycerides (TG) between male and female offspring. Sex-specific differences separated by dietary treatment group for (**J**) diglycerides (DG; water-loss product from fragmentation in source), (**K**) phosphatidylcholine (PC), (**L**) phosphatidylethanolamines (PE), (**M**) sphingomyelins (SM) and (**N**) triglycerides (TG). N = 20 offspring (10 males and females) per treatment group, sampled from all litters generated. Data are expressed as mean ± SEM (**A**,**B**,**D**–**M**). * *p* < 0.05, ** *p* < 0.01, *** *p* < 0.001. Statistical differences were determined using a random effects regression analysis (**A**,**B**), one-way ANOVA with Bonferroni post hoc correction (**C**) or Student’s *t*-test (**D**–**M**).

**Figure 4 ijms-24-01814-f004:**
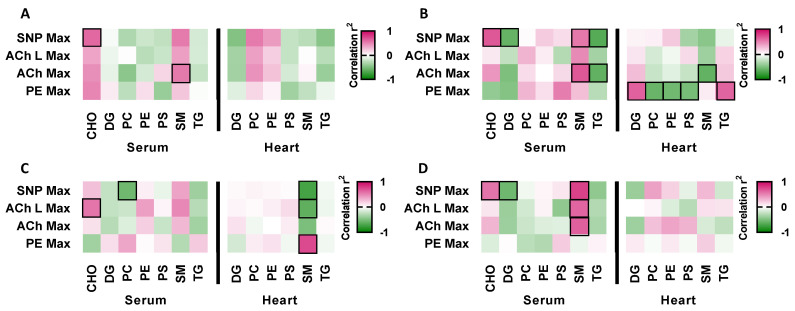
Correlations between serum, heart and brain lipids in F1 adult offspring and vascular function parameters: Correlation between previously reported maximal (max) mesenteric artery responses to phenylephrine (PE Max), acetylcholine (Ach Max), Acetylcholine with Nώ-nitro-Larginine methyl ester (L-NAME; ACh L Max) or sodium nitroprusside (SNP Max) with a relative Z-score abundance of serum cholesterol (SHO), diglycerides (SDG; water-loss product from fragmentation in source), phosphatidylcholines (SPC), phosphatidylcholines (SPE), phosphatidylserine (SPS), sphingomyelins (SSM) and triglycerides (STG) as well as heart diglycerides (HDG; water-loss product from fragmentation in source), phosphatidylcholines (HPC), phosphatidylcholines (HPE), phosphatidylserine (HPS), sphingomyelins (HSM) triglycerides (HTG) and cerebrum diglycerides (CDG), phosphatidylcholines (CPC), phosphatidylcholines (CPE), phosphatidylserine (CPS), sphingomyelins (CSM) and triglycerides (CTG). Correlations are reported in (**A**) NN (NPD sperm and LPD seminal plasma), (**B**) LL (LPD sperm and LPD seminal plasma), (**C**) NL (NPD sperm and LPD seminal plasma) and (**D**) LN (LPD sperm and NPD seminal plasma) adult offspring tissues. *N* = 10–20 offspring (5–10 males and females) per treatment group, sampled from all litters generated. Statistical differences were determined using Pearson’s correlation with significance taken at *p* < 0.05. Black boxes indicate statistically significant correlations.

**Figure 5 ijms-24-01814-f005:**
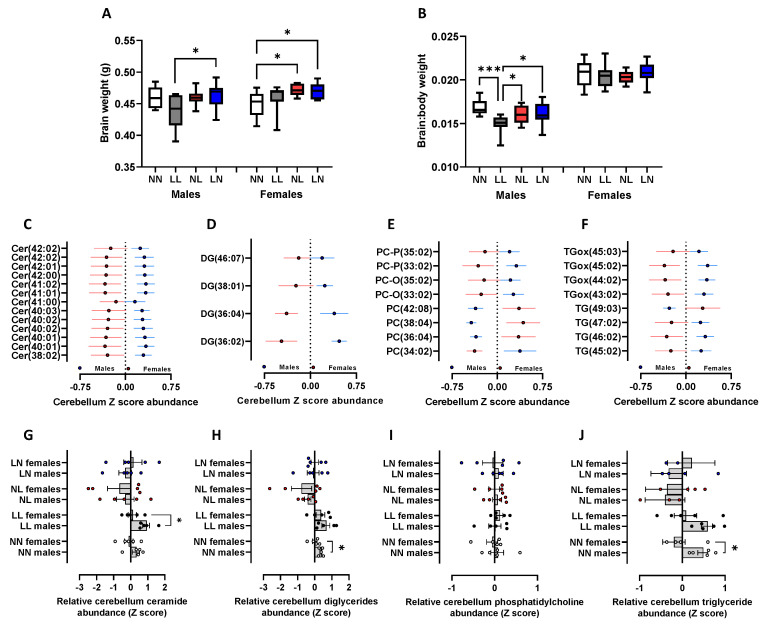
Impact of paternal diet on F1 adult offspring brain tissue lipid profiles: (**A**) the brain weight and (**B**) brain:body weight ratio of NN (NPD sperm and LPD seminal plasma), LL (LPD sperm and LPD seminal plasma), NL (NPD sperm and LPD seminal plasma) and LN (LPD sperm and NPD seminal plasma) offspring at 16 weeks of age. Sex-specific differences in cerebellum (**C**) ceramides, (**D**) diglycerides (DG; water-loss product from fragmentation in source), (**E**) phosphatidylcholine (PC) and (**F**) triglycerides (TG) between male and female offspring. Sex-specific differences separated by dietary treatment group for (**G**) ceramides, (**H**) diglycerides (DG), (**I**) phosphatidylcholine (PC) and (**J**) triglycerides (TG). N = 10 offspring (five males and females) per treatment group, sampled from all litters generated. Data are expressed as mean ± SEM. * *p* < 0.05, *** *p* < 0.001. Statistical differences were determined using a random effects regression analysis (**A**,**B**) or Student’s *t*-test (**C**–**J**).

**Figure 6 ijms-24-01814-f006:**
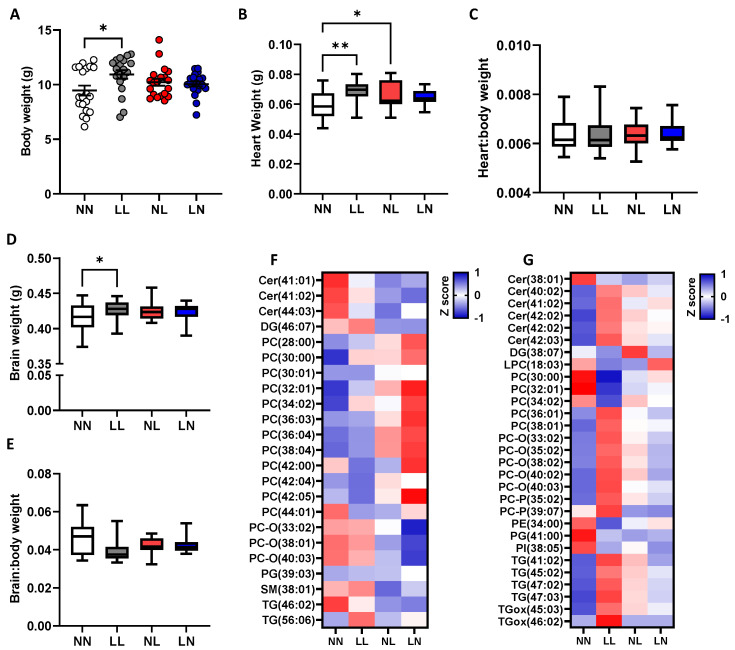
Impact of paternal diet on F2 neonatal offspring heart and brain tissue lipid profiles: (**A**) body weight, (**B**) heart weight, (**C**) heart:body weight ratio, (**D**) brain weight and (**E**) brain:body weight ratio of NN (NPD sperm and LPD seminal plasma), LL (LPD sperm and LPD seminal plasma), NL (NPD sperm and LPD seminal plasma) and LN (LPD sperm and NPD seminal plasma) F2 offspring at 3 weeks of age. Z-scores of differential lipids between NN, LL, NL and LN offspring (**F**) cerebellum and (**G**) cerebrum. *N* = 10–20 offspring (5–10 males and females) per treatment group, sampled from all litters generated. Data are expressed as mean ± SEM (**A**–**D**). * *p* < 0.05, ** *p* < 0.01. Statistical differences were determined using a random effects regression analysis (**A**–**D**) or one-way ANOVA with Bonferroni post hoc correction (**E**,**F**).

## Data Availability

All data underlying the graphs and charts presented in the main figures can be found within in the Appendix A or are available on request from the corresponding author.

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
