# Peer review of "Characterisation of the Paternal Influence on Intergenerational Offspring Cardiac and Brain Lipid Homeostasis in Mice"

_ijms, 2023, doi:10.3390/ijms24031814_

Round 1
Reviewer 1 Report
This is an interesting paper by Furse et al, looking at the role of paternal environment, particularly paternal diet around conception and the implications of the offspring and their offspring.
The authors may wish to consider the following points.
Materials and Methods
line 668. This implies that the absence of a vaginal plug means unsuccessful mating - were they removed or was mating continued for subsequent nights?
Line 698. When the instrument was available? Not sure what is meant by this. If it is simply a logistical concern of the local lab it is probably unnecessary here.
Results
Section 2.1 - maybe it is a journal style issue but the opening section of results runs from line 95 to 141 in a single, unbroken paragraph. Given that it starts with some context about the model it could do with being broken up somewhat to make clear where the actual results begin. Personally I'd prefer to see context removed from the results and placed into the discussion or introduction and leave the results section for what these experiments have produced.
Figures 1 and 2: it might be useful to have body weights of the groups presented somewhere. The NL group is showing raised heart and brain to BW ratio despite no change or small decrease in size of the organ. From this I would guess that this group has a smaller (perhaps not significantly) body weight.
Stats. From the stats in the Figure legend, the regression model was used for the organ weight/body weight data but a One way ANOVA for the lipid comparison? Does it not fit the model? Also, since it is a diet in semen vs. diet in seminal fluid is it not a 2x2 design and so a two-way analysis of variance?
line 294 and vasodilatory experiments. I realise that this is previous work but also the experiments and publishing don't fall on the same time frame. Can you confirm if this data relates to the same animals. A given mouse's cholesterol abundance is related to it's measure vascular function?
line 308 & 317 ACh
Discussion + general points
Is there an overall pattern forming, an enrichment of one species or group (ceramides, PC etc) over time in one place? If so it didn't come across in the manuscript.
Do you know what the liver or circulation is doing in relationship to these fats? Do they reflect broader changes elsewhere?
How do the F2 profiles in a given sex/group compare to the corresponding F1 profiles, are they similar? That is to say does the NN F2 look like the NN F1? Are they converging towards more homogeneity or diverging into more variability?
Author Response
Manuscript: ijms-2052165
Characterisation of the paternal influence on intergenerational offspring cardiac and brain lipid homeostasis in mice
Reviewer 1
This is an interesting paper by Furse et al, looking at the role of paternal environment, particularly paternal diet around conception and the implications of the offspring and their offspring. The authors may wish to consider the following points.
Response: We thank the Reviewer for taking the time to consider our manuscript and for their interest in the study. We have addressed their comments and we hope this has improved the manuscript.
Materials and Methods
line 668. This implies that the absence of a vaginal plug means unsuccessful mating - were they removed or was mating continued for subsequent nights?
Response: We thank the reviewer for their comment on this. As the females were superovulated, mating occurred in the majority (> 90%) of times they were paired with males. However, although the females were mated by our vasectomised males, they did not always become pregnant. Females were monitored for signs of weight gain as an indicator of pregnancy. Females who did not become pregnant were removed from the study and were not used again. We have amended the text to make this clearer, please see lined 672-674:-
‘Females were monitored for signs of weight gain, indicating pregnancy. Females who did not become pregnant were removed from the study and were not mated again’.
Line 698. When the instrument was available? Not sure what is meant by this. If it is simply a logistical concern of the local lab it is probably unnecessary here.
Response: We agree with the reviewer that this information is not necessary and so have removed this statement. We have also slightly revised this sentence, please see lines 702-705:-
‘When enough samples to fill 4 × 96 well plates had been processed, the dried films were re-dissolved (tert-butylmethyl ether, 20 µL/well; MS-mix, 80 µL/well) and the plate was heat-sealed and run immediately, with the first injection within 10 minutes’.
Results
Section 2.1 - maybe it is a journal style issue but the opening section of results runs from line 95 to 141 in a single, unbroken paragraph. Given that it starts with some context about the model it could do with being broken up somewhat to make clear where the actual results begin. Personally I'd prefer to see context removed from the results and placed into the discussion or introduction and leave the results section for what these experiments have produced.
Response: We decided to place some context into the results section as they come before the Materials and Methods and so the reader may not be familiar with the nature of our model. As we have generated a relatively complex series of offspring mice, we felt it necessary to explain what the 4 groups (NN, LL, NL and LN) were. However, we appreciate that the details regarding our previous studies could be removed from the opening paragraph of the Results, as these are covered elsewhere in the manuscript. This had reduced the length of this paragraph.
Figures 1 and 2: it might be useful to have body weights of the groups presented somewhere. The NL group is showing raised heart and brain to BW ratio despite no change or small decrease in size of the organ. From this I would guess that this group has a smaller (perhaps not significantly) body weight.
Response: As suggested, we have now added in body weight data to Figure 1 (panel A), Figure 3 (panel A) and figure 6 (Panel A).
Stats. From the stats in the Figure legend, the regression model was used for the organ weight/body weight data but a One way ANOVA for the lipid comparison? Does it not fit the model? Also, since it is a diet in semen vs. diet in seminal fluid is it not a 2x2 design and so a two-way analysis of variance?
Response: We thank the Reviewer for the comments regarding the statistical design. Where we had more than one male and/or female from each litter (as in the data on body weights and organ weights) we used a liner regression to account for random factors such as litter size. For the analysis of tissue lipids, we only took one male and one female from each group and so it was not possible to account for ‘litter’ effects within this data. As such, a One-Way ANOVA approach was adopted.
We are currently looking to mathematically model the data presented within this study, as well as our recently published paper (Morgan et al. Paternal low protein diet perturbs inter-generational metabolic homeostasis in a tissue-specific manner in mice. Commun Biol. 2022. 5(1):929) to get a better understanding of how paternal diet affects tissue lipid abundance in sperm/seminal plasma-, age and sex-specific mechanisms.
line 294 and vasodilatory experiments. I realise that this is previous work but also the experiments and publishing don't fall on the same time frame. Can you confirm if this data relates to the same animals. A given mouse's cholesterol abundance is related to it's measure vascular function?
Response: We can confirm that the samples analysed in this study were from the same mice as those analysed in our previous study (Morgan et al. Paternal diet impairs F1 and F2 offspring vascular function through sperm and seminal plasma specific mechanisms in mice.
J Physiol. 2020. 598(4):699-715). As such, we are able to correlate directly the vascular responses in these mice with their tissue lipid profiles. We do indicate (line 286) that these are the same animals as previously published.
line 308 & 317 Ach
Response: This has now been corrected, see lines 297 and 306.
Discussion + general points
Is there an overall pattern forming, an enrichment of one species or group (ceramides, PC etc) over time in one place? If so it didn't come across in the manuscript.
Response: We agree with the Reviewer that attempting to understand the patterns of lipid abundance changes across tissues, sexes and ages would be of significant interest. One issue with such an interpretation is that our abundance data are relative. As such, we are unable to state whether changes are occurring due to an increase in one group or a decrease in another. Therefore we are currently unable to comment on absolute changes in total lipid content and abundance over time.
Do you know what the liver or circulation is doing in relationship to these fats? Do they reflect broader changes elsewhere?
Response: We have recently published the details of the lipid profiles in the liver, adipose and serum from these same animals (Morgan et al. Paternal low protein diet perturbs inter-generational metabolic homeostasis in a tissue-specific manner in mice. Commun Biol. 2022. 5(1):929). We have also begun to construct models that assess the distribution of lipids within and between the tissues (Furse et al. Lipid Traffic Analysis reveals the impact of high paternal carbohydrate intake on offsprings' lipid metabolism. Commun Biol. 2021 Feb 5;4(1):163). What is interesting is that we see a range of tissue, age and sex specific responses rather than a broad overarching pattern. This may reflect the differences between storage (adipose), metabolic (liver), functional (heart) and the brain tissues. We are looking to develop a more comprehensive mathematical model which integrates all the different tissues, ages and both sexes to define how tissue lipid abundance changes. We have added some more text to the Discussion highlighting the need for such a model and the comparisons between this study and our previously published work. See lines 628-635:-
‘Interestingly, we observed a greater number of individual lipids to have a differential abundance in F2 neonatal hearts and brains that in our F1 neonates. Previously, we have shown that in the livers of the same animals as analysed here, the ratio of phosphatidylcholine to phosphatidylethanolamine (PC:PE) in NL and LN offspring was more similar to the F1 adults than the F1 neonates [29]. While these data indicate a drift in tissue lipid profile across generations, more complex modelling of the lipid abundance between tissues, sexes, ages and generations is required to fully understand intergenerational paternal programming’.
How do the F2 profiles in a given sex/group compare to the corresponding F1 profiles, are they similar? That is to say does the NN F2 look like the NN F1? Are they converging towards more homogeneity or diverging into more variability?
Response: We thank the Reviewer for raising this interesting question. In our previous publication (Morgan et al. Paternal low protein diet perturbs inter-generational metabolic homeostasis in a tissue-specific manner in mice. Commun Biol. 2022. 5(1):929) we saw that in the F2 neonate liver, ratio of phosphatidylcholine to phosphatidylethanolamine was more similar to that seen in the F1 adults than the F1 neonates, suggesting that the F2 neonates may be more similar to that of an adult.
In the current study, while similar phosphatidylcholine:phosphatidylethanolamine patterns were not observed, there were more individual lipids displaying a significant difference between groups in the F2 neonates than in the F1 neonates. This might suggest a more heterogeneous population, the significance of these changes remains to be defined. However, we have added in more text to highlight the need for more detailed studies, please see lines 628-635:-
‘Interestingly, we observed a greater number of individual lipids to have a differential abundance in F2 neonatal hearts and brains that in our F1 neonates. Previously, we have shown that in the livers of the same animals as analysed here, the ratio of phosphatidylcholine to phosphatidylethanolamine (PC:PE) in NL and LN offspring was more similar to the F1 adults than the F1 neonates [29]. While these data indicate a drift in tissue lipid profile across generations, more complex modelling of the lipid abundance between tissues, sexes, ages and generations is required to fully understand intergenerational paternal programming’.
Reviewer 2 Report
This is an interesting and well-carried-out article, however, some points could be improved:
1.- In the abstract, results and conclusion should be more detailed and indicate the relevance of sperm and seminal fluid on the outcome. Also, some indication of the meaning of the changes in the lipid profiles should be incorporated (not only that they change).
2.- The data is overwhelming, and at some point to clarify the meaning of the results I would suggest the incorporation of ratios. For example the correlation among groups in fold-change with TAG carbon number, and with acyl chain double bonds, as well as relevant lipid species and ratios: TAGs enriched in saturated fatty acids, TAGs enriched in polyunsaturated fatty acids, lipogenic index (The ratio of FA (16:0/18:2n-6) and the ratio Docosahexaenoyl Phosphatidylcholine/Phosphatidylcholine (PC–DHA/PC).
Author Response
Manuscript: ijms-2052165
Characterisation of the paternal influence on intergenerational offspring cardiac and brain lipid homeostasis in mice
Reviewer 2
This is an interesting and well-carried-out article, however, some points could be improved:
1.- In the abstract, results and conclusion should be more detailed and indicate the relevance of sperm and seminal fluid on the outcome. Also, some indication of the meaning of the changes in the lipid profiles should be incorporated (not only that they change).
Response: We thank the Reviewer for their time and consideration of our manuscript. We acknowledge that our study has not explored the physiological or health consequences of the changes that are being described in this study. We would like to highlight that we have previously published detailed cardiovascular analysis of these same offspring (Morgan et al., Paternal diet impairs F1 and F2 offspring vascular function through sperm and seminal plasma specific mechanisms in mice. J Physiol. 2020. 598(4):699-715). We have also recently published detailed insight to the central metabolic status of these mice (Morgan et al. Paternal low protein diet perturbs inter-generational metabolic homeostasis in a tissue-specific manner in mice. Commun Biol. 2022. 5(1):929). We would also like to highlight that within these, as well as other associated papers from the same model (Watkins et al. Paternal diet programs offspring health through sperm- and seminal plasma-specific pathways in mice.
Proc Natl Acad Sci U S A. 2018. 115(40):10064-10069), we have explored our hypothesis on the seminal and sperm specific mechanisms of paternal programming.
However, we acknowledge that for readers who are accessing this study without prior knowledge of our previous work a discussion on the relevance of sperm and seminal fluid would be of benefit. In the Introduction (lines 75-82) give details on the specific sperm and seminal plasma mechanisms that are currently known about. In addition, in the Discussion (lines 640-645) we have added extra text covering the role that the seminal plasma may play in modulating the maternal environment and so affecting embryonic and fetal development:-
‘Additionally, our existing, and previous, data show that the seminal plasma also plays a significant role in influencing offspring phenotype and life-long wellbeing. These seminal plasma effects are likely mediated through responses in the maternal uterine immune, vascular and inflammatory status during the earliest stages of development [27]. Together, these data highlight the mechanistic complexity of paternal programming’.
While we acknowledge that our study is largely descriptive in nature, we believe we have tried to relate changes in lipid abundance with cardiac and cognitive function, as well as highlighting the sex-specific differences. For example in the Discussion (lines 488-496) we state:-
‘Accumulation of metabolic intermediates, such as diglycerides and triglycerides in the heart has been linked with insulin resistance and a range of cellular perturbations including apoptosis and endoplasmic reticulum stress and oxidative stress [39]. In skeletal muscle, intramyocellular diglyceride accumulation has been linked with insulin resistance, blunted insulin signalling and lipotoxicity [40]. However, some studies have indicated that streptozotocin-induced diabetic myocardial dysfunction can be improved via increasing the levels of cardiac diglycerides and fatty acids in rats [41]’.
In the following section (lines 515-527 of the Discussion) we link brain lipids to cognitive function and the role of parental diet.
Furthermore, in the following section we then discuss the relative sex-specific differences in tissue lipid abundance and relate these to the know differences in incidences of cardiovascular diseases seen between men and women. As such, we do believe that we have placed the tissue and sex-specific lipid abundance changes observed in our study into context. However, the underlying mechanisms through which paternal diet programs offspring lipid metabolism are still to be defined.
2.- The data is overwhelming, and at some point to clarify the meaning of the results I would suggest the incorporation of ratios. For example the correlation among groups in fold-change with TAG carbon number, and with acyl chain double bonds, as well as relevant lipid species and ratios: TAGs enriched in saturated fatty acids, TAGs enriched in polyunsaturated fatty acids, lipogenic index (The ratio of FA (16:0/18:2n-6) and the ratio Docosahexaenoyl Phosphatidylcholine/Phosphatidylcholine (PC–DHA/PC).
Response: we acknowledge the Reviewer’s comment regarding making the data as accessible to the reader as possible. We would like to highlight that we have tried to simplify the data providing an overview of the main lipid structures (i.e. Figure 1D showing percentage saturated, mono-unsaturated and poly-unsaturated), as well as then broken into the main lipid classes (e.g Figure 1E showing % of ceramides, diglycerides, phosphatidylcholine, phosphatidylethanolamine, triglycerides, etc.). However, such diagrams have only been used where differences in these groups were observed. For the F1 adults, sex was more of an influence than diet group alone and so we have shown the abundance of the main lipid classes with regard to sex and diet. In the F2 offspring, minimal lipid class-specific changes were observed and so we have presented the data based on the different individual lipids.
We thank the Reviewer for their suggestion on other lipid correlations and ratios to analyse within our data. However, there are very many different ratio’s that could be tested and would even further expand the amount of data to cover within this manuscript. For example, we have compared the relative abundance of saturated, mono-unsaturated and poly-unsaturated TGs in our offspring tissues. As shown below, there are no differences in the relative levels in either the hearts or brains of our F1 adults. This was also the case for the F1 neonates and the F2 neonates (data not shown). As such, we have decided not to include such additional analyses in our manuscript. In addition, we compared the ratio tissue PC:PE, a known regulator of cell membrane integrity and plays a role in the progression of steatosis into steatohepatitis. However, we observed no difference in this ratio across groups, tissues or sexes.

Reviewer 3 Report
The authors report here the effects of a paternal low protein diet on the lipid content in heart and brain of the offspring. The manuscript is well written and the data clearly presented, with an exhaustive review of the lipid content in brain and heart. Although the mechanisms are not investigated in this paper, the data presented here are of interest. It is now evident that paternal environment can influence offspring's health, and the data presented here are of importance in the field.
I therefore recommend acceptance of the paper in the present form.
Author Response
We thank the Reviewer for their time and consideration of our manuscript. We appreciate the Reviewer’s comment on the lack of mechanism and this is something we are actively seeking to explore in subsequent studies. We also thank the Reviewer for their recognition of the importance of our paper and its findings.
Round 2
Reviewer 1 Report
The authors have addressed all comments